# Co-circulation of multiple influenza A reassortants in swine harboring genes from seasonal human and swine influenza viruses

Pia Ryt-Hansen[1,2]\*, Jesper Schak Krog[3], Solvej Østergaard Breum[3], Charlotte Kristiane Hjulsager[3], Anders Gorm Pedersen[4], Ramona Trebbien[3], Lars Erik Larsen[1,2]

[1]Technical University of Denmark, National Veterinary Institute, Lyngby, Denmark; [2]University of Copenhagen, Department of Health Sciences, Institute for Animal and Veterinary Sciences, Frederiksberg, Denmark; [3]Statens Serum Institut, Copenhagen, Denmark; [4]Department of Health Technology, Section for Bioinformatics, Technical University of Denmark, Kongens Lyngby, Denmark

**Abstract** Since the influenza pandemic in 2009, there has been an increased focus on swine influenza A virus (swIAV) surveillance. This paper describes the results of the surveillance of swIAV in Danish swine from 2011 to 2018. In total, 3800 submissions were received with a steady increase in swIAV-positive submissions, reaching 56% in 2018. Full-genome sequences were obtained from 129 swIAV-positive samples. Altogether, 17 different circulating genotypes were identified including six novel reassortants harboring human seasonal IAV gene segments. The phylogenetic analysis revealed substantial genetic drift and also evidence of positive selection occurring mainly in antigenic sites of the hemagglutinin protein and confirmed the presence of a swine divergent cluster among the H1pdm09Nx (clade 1A.3.3.2) viruses. The results provide essential data for the control of swIAV in pigs and emphasize the importance of contemporary surveillance for discovering novel swIAV strains posing a potential threat to the human population.

\*For correspondence:
piarh@sund.ku.dk

Competing interests: The authors declare that no competing interests exist.

## Introduction

Influenza A virus (swIAV) infection in swine causes respiratory disease, impairs the growth rate, and increases the risk of secondary infections (*Opriessnig et al., 2011*; *Brown et al., 1993*; *Jung et al., 2005*). SwIAV is enzootic globally, and multiple subtypes and lineages have been identified (*Vincent et al., 2014*). The influenza A virus (IAV) genome consists of eight distinct gene segments, and subtypes are assigned by characterizing the two surface glycoproteins hemagglutinin (HA) and neuraminidase (NA) (*Medina and García-Sastre, 2011*).

Pigs are infected by the same subtypes as humans, including H1N1, H1N2, and H3N2 (*Vincent et al., 2014*). The transmission of H1N1 avian IAV to swine in the 1970s created the Eurasian avian-like swine H1N1 lineage circulating in Europe and Asia (*Pensaert et al., 1981*). This lineage was later defined as the 1.C lineage and will be termed H1avN1av in this study (*Anderson et al., 2016*). An H3N2 influenza virus related to a human strain from 1973 started to circulate in the European pig populations in 1984. In the mid-1980s, a reassortment between the H1avN1av and H3N2 human virus resulted in a human-like reassortant swine 'H3N2' that became established in European swine (*Castrucci et al., 1993*; *Haesebrouck et al., 1985*). In 1994, a H1N2 reassortant (1.B lineage) comprising an HA gene from human seasonal H1N1, an NA gene from H3N2, and internal genes originating from H1avN1av was first identified in the United Kingdom and

subsequently detected in many European countries (*Brown et al., 1998*). This swIAV lineage is also known as European human-like 'H1huN2.' However, this lineage has never been detected in Danish pigs. In the beginning of the 2000s, a new H1N2 reassortant virus was identified in Danish pigs (*Trebbien et al., 2013*). This 'H1N2dk' virus comprised an HA gene of the H1avN1av lineage and an NA from the contemporary, circulating H3N2 and has since been identified in several European countries (*Bálint et al., 2009*, *Moreno et al., 2012*, *Watson et al., 2015*, *Simon et al., 2014*).

In 2009, a novel IAV identified as pandemic H1N1/2009 strain of influenza A (1A.3.3.2 lineage – H1N1pdm09) spread rapidly among humans worldwide. The H1N1pdm09 virus is a reassortant, which obtained most of its gene segments from the triple-reassortant swIAV circulating in North American swine, its NA and matrix (M) gene segments from the Eurasian avian-like swine H1N1 lineage (*Garten et al., 2009*; *Smith et al., 2009*) and had its origin in the Mexican swine population (*Mena et al., 2016*). Soon after the virus began to spread globally in humans, its introduction into the swine population was noticed in several countries (*Watson et al., 2015*; *Forgie et al., 2011*; *Welsh et al., 2010*; *Matsuu et al., 2012*). In transmission experiments, the high susceptibility of pigs to H1N1pdm09 infection was confirmed as well as an efficient pig-to-pig transmission (*Brookes et al., 2010*). This instantly raised concerns about the possible generation of new reassortants between H1N1pdm09 virus and circulating swIAV lineages, which soon after was indeed found to have occurred in several countries, including Denmark (*Watson et al., 2015*; *Howard et al., 2011*; *Vijaykrishna et al., 2010*; *Krog et al., 2017*; *Starick et al., 2012*). Consequently, there is a risk for the development of novel and more virulent progeny virus capable of infecting humans.

Surveillance of swIAV in pigs concerns both animal and public health. For animal health, the documentation of enzootic and new emerging swIAV and their ecology is important for control of disease and to ensure the use of adequate diagnostic tools. From a public health point of view, the results are important for risk assessments of emerging IAV, resistance to antiviral drugs, or increased pathogenicity as well as pandemic preparedness. Here we report the results of a passive surveillance program of swIAV conducted in Denmark from 2011 to 2018, including data on intensive genetic characterization of swIAV-positive submissions.

## Results

### Field samples

The total number of submissions received for swIAV diagnostics from pigs with acute respiratory disease in the years 2011–2018 fluctuated over the years, with a peak in 2015 (*Figure 1—figure supplement 1*). In total, 3800 submissions were received over 8 years. A *submission* to the surveillance program was defined as a number of sample from a single herd at a certain date. The pattern of monthly submissions was very similar from year to year showing a peak in the number of submissions from October to March (autumn and winter months) (*Figure 1—figure supplement 1*). When comparing the number of swine herds submitting samples for swIAV diagnostics each year (n = 276–488) with the total number of swine herds present in Denmark the same year (n = 2741–4529) (*Christiansen, 2019*), it was evident that 6–15% of the Danish swine herds were included in the surveillance with a steady increase over the years. The calculation was corrected for repeated submissions from the same herd. The first year of the surveillance, 36% of the total submissions contained at least one positive sample. The following five years (2012–2016), the percentage of swIAV-positive submissions was stable ranging between 44 and 47%, whereas a marked increase was observed over the last two years, reaching 56% in 2018 (*Figure 1a*). The monthly distribution of swIAV-positive submissions was fluctuating, but no consistent seasonal variations were observed (*Figure 1—figure supplement 2*). The average monthly percentage of positive submission over the 8 years ranged from 42.6 to 51.8% with the highest average percentages in April, September, and December.

### Test for the HA gene of H1N1pdm09 origin by specific real-time PCR

Due to the global spread of H1N1pmd09 virus in humans, it was decided in 2011 to test all swIAV-positive samples from Danish pigs specifically for the presence of the HA gene of H1N1pdm09 origin (H1pdm09). In 2011, 21% of the swIAV-positive submissions tested positive for H1pdm09. However, in the two following years the percentage decreased to 14–16% of the swIAV-positive submissions. This decrease reverted in 2014, where a marked increase was observed, and since

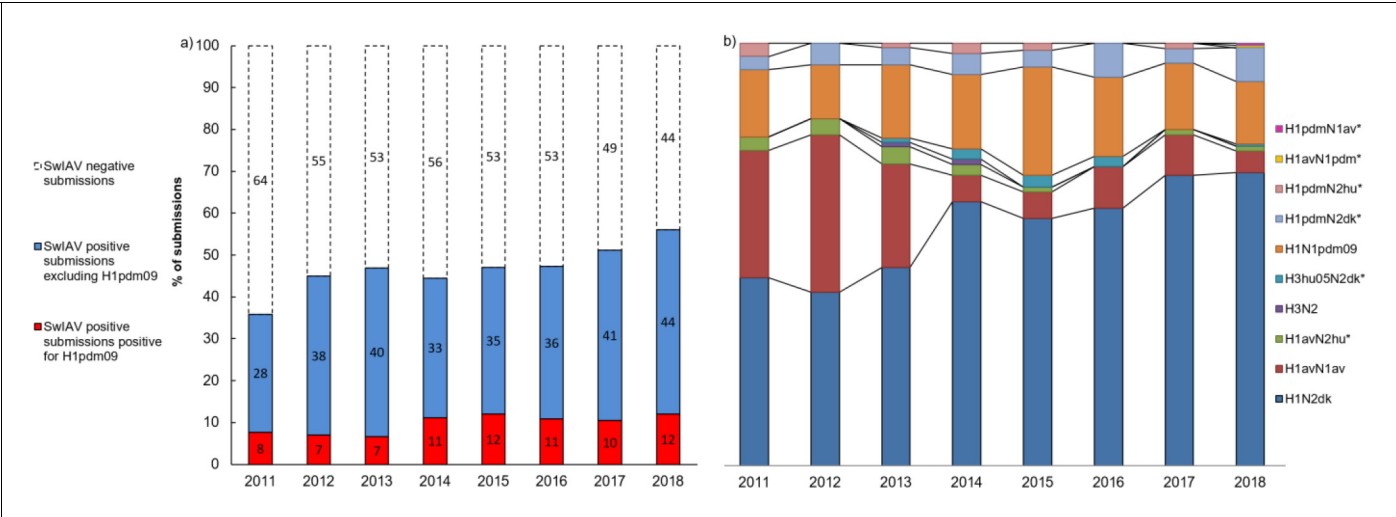

**Figure 1.** Submission and lineage distribution 2011-2018. (**a**) The percentages of submissions testing negative and positive for influenza A virus including the proportion of positive submissions containing a hemagglutinin (HA) of H1N1pmd09 origin. (**b**) Lineage distribution of the influenza A-positive samples. * indicates the novel reassortants discovered during the surveillance.

The online version of this article includes the following figure supplement(s) for figure 1:

**Figure supplement 1.** The annual and monthly number of submissions received from Danish pigs with acute respiratory disease in the years 2011–2018.

**Figure supplement 2.** Monthly distribution of submissions for the Danish swine influenza A virus (swIAV) surveillance 2011–2018.

then, the proportion of H1pdm09 remained at a stable level, ranging between 20 and 26% of the swIAV-positive submissions (*Figure 1a*). On average, H1pdm09-positive submissions constituted 15–25% of the monthly swIAV-positive submissions (*Figure 1—figure supplement 2*). No significant differences (p value>0.05) in the average proportions of H1pdm09-positive submissions between the different months were observed.

## SwIAV lineages

During the years 2011–2014, 33–48% of the swIAV-positive submissions categorized into lineages partial sequencing of the HA and NA surface genes. From 2015 and onward, the lineage of the swIAV-positive submissions was determined by multiplex RT-PCR and/or by Fluidigm, resulting in an increase in successful lineage determination corresponding to 61–77% of the total number of swIAV-positive submissions (data not shown). Mixed infections were rarely observed and only submissions classified as containing a single lineage were included in this study.

The most prevalent lineage identified in the swIAV-positive submissions during the 8 years (2011–2018) was H1N2dk containing the HA of the Eurasian avian-like swine H1N1 lineage (H1av). The proportion of H1N2dk lineage increased steadily from 42% of the positive submissions in 2012 to 69% of the positive submissions in 2018. In contrast, the H1avN1av, which was highly prevalent in 2011–2012 representing between 30 and 37% of the positive submissions, decreased markedly since then, only representing 5% of the swIAV-positive submissions in 2018 (*Figure 1b*).

The proportion of H1N1pdm09 was relatively stable from 2011 to 2018 representing approximately 16% of the positive submissions with the exception of 2015 (25.6 %). Several reassortants containing either the HA or the NA gene of the H1N1pdm09 lineage were identified. The most prevalent of these reassortants was the 'H1pdmN2dk,' which combined the HA gene of the H1N1pdm09 lineage and the NA gene of the H1N2dk lineage. This reassortant was detected for the first time in Denmark in 2011, and since then, the proportion of this reassortant remained relatively stable constituting around 4% of the positive submissions each year. However, in the years 2016 and 2018, a doubling in prevalence of H1pdmN2dk was observed. Another reassortant, also detected for the first time in 2011, was the 'H1pdmN2hu95,' which contained an HA gene of H1N1pdm09 origin and an NA gene derived from the human seasonal flu circulating in 1995. This reassortant was identified with low prevalence (1–3.2%) from 2011 to 2017, but was interestingly not detected in the years

where the prevalence of H1pdmN2dk peaked (2016 and 2018). In 2018, two novel swIAV reassortants were identified, both including one surface gene of H1N1pdm09 origin. One was termed 'H1avN1pdm' and had an avian-like swine HA gene and an NA gene of H1N1pdm09 origin. The other novel reassortant was termed 'H1pdmN1av' and carried an HA gene of H1N1pdm09 origin and an NA gene derived from the H1avN1av (*Figure 1b*).

The swine-adapted reassortant H3N2 was detected in a few samples in 2013–2014, but was not detected in 2015–2018. However, another H3N2 reassortant 'H3hu05N2dk,' containing an HA gene of human seasonal origin from 2005 and an NA gene of the H1N2dk lineage, has been detected each year since 2013, with the exception of 2017 (*Krog et al., 2017*). Another reassortant, containing the N2 gene of the human seasonal H3N2 lineage, was detected for the first time in Denmark in 2011 and was termed 'H1avN2hu95' (*Breum et al., 2013*). This lineage carried an avian-like swine HA gene and an NA gene derived from the human seasonal flu circulating in 1995. The H1avN2hu95 lineage has been detected each year, with the exception of 2016 (*Figure 1b*).

In summary, six novel swIAV reassortant strains (H1pdmN2dk, H1pdmN2hu95, H1pdmN1av, H1avN1pdm, H3hu05N2dk, and H1avN2hu95) were discovered through the Danish surveillance of swIAV from 2011 to 2018 (*Figure 1b*). However, the diversity of circulating strains is even more complex, when all gene segments are included in the analyses as described below.

## Full-genome sequencing

In total, 128 full-genome sequences of swIAV isolated between 2013 and 2018 were uploaded in NCBI GenBank with the following accession numbers: MT666225–MT667233. The accessions numbers, corresponding sample IDs, and information on the lineage of each gene segments are summarized in *Supplementary file 1*. The characteristics of the H3hu05N2dk virus (A/swine/Denmark/2014_15164_1p1/2014(H3N2) accession number: EPI_ISL_247092) have previously been described (*Krog et al., 2017*), but were also included in the analysis of the H3hu genes of this study, resulting in 129 sequences used for the following analysis.

## Hemagglutinin gene characterization

In total, 78 H1av, 48 H1pdm09, and 3 H3hu05 full-length HA sequences were obtained and analyzed separately according to the lineage.

The H1av nucleotide sequences showed an average pairwise sequence difference per site (pi) of 0.099, SE 0.0005, indicating a high-sequence diversity. The phylogenetic analysis of the H1av sequences revealed that they did not display the imbalanced, ladder-like structure typical for influenza trees (*Figure 2*), and there was a low correlation between sampling time and genetic divergence (*Figure 2—figure supplement 1* and *Table 1*). Conversely, the H1av sequences were dispersed into several clusters (*Figure 2*), with all but one cluster containing at least one sequence representative of the 10 European H1av clusters recently defined by *Henritzi et al., 2020*. One cluster, however, with a low posterior possibility (0.6), contained only Danish sequences and did not show a close sequence identity to other H1av foreign sequences in NCBI GenBank. Another interesting observation was the presence of one major cluster, which was dominated by H1avNx strains carrying a complete internal gene cassette of H1N1pdm09 origin. The global references obtained from IRD (http://www.fludb.org) to define the different H1av clades (1.C.1, 1.C.2, 1.C.2., 1.C.2.2., and 1.C.2.3) (*Anderson et al., 2016*) did not cluster closely together with the majority of the Danish H1av sequences. However, all the Danish H1av of this study did belong to the 1.C.2. clade when using the 'Swine H1 Clade Classification Tool' offered by IRD. This could indicate that the IRD classification of the H1av clades needs to be expanded. The HA nucleotide sequence obtained from the novel G4 swIAV found in China (*Sun et al., 2020*), carrying a Eurasian avian like HA gene of the 1.C.2.3. clade, was included in the phylogenetic analysis and did not cluster close to any of the Danish H1av sequences. Analysis using CODEML indicated strong evidence for positive selection among the H1av sequences. Specifically, the dN/dS ratios for individual codons under the M2a model strongly suggested the presence of positive selection in four positions, all located in the globular head of the HA protein and all in previously defined antigenic sites (*Table 1*).

The H1pdm09 nucleotide sequences had a lower nucleotide diversity, pi = 0.043, SE 0.0005, compared to the H1av sequences. The phylogenetic analysis of the H1pdm09 sequences showed that 30 of the Danish H1pdm09 sequences of this study were located in a well-defined cluster along with 16

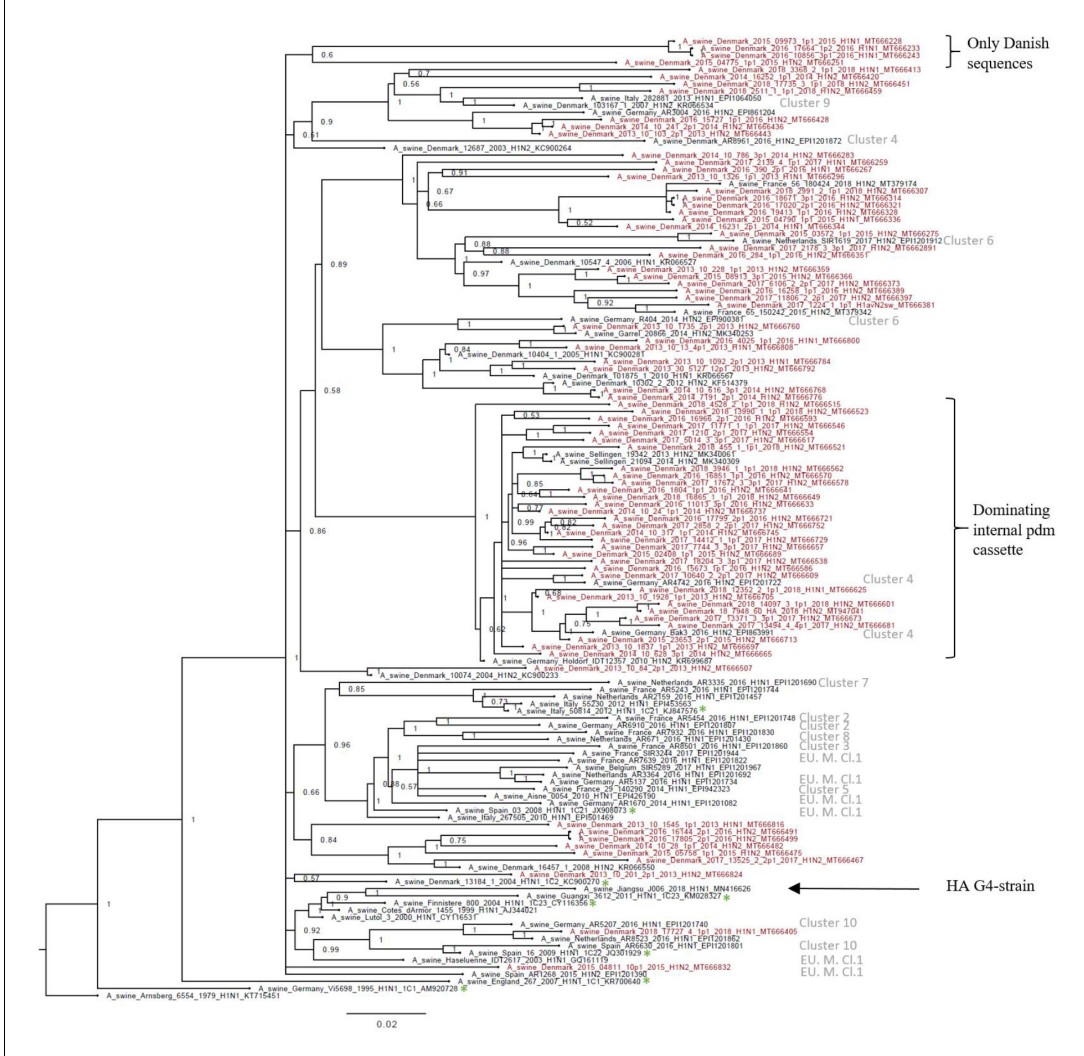

**Figure 2.** Bayesian phylogenetic tree of the Danish hemagglutinin (HA) nucleotide sequences of H1av origin from 2011 to 2018 and HA reference sequences of H1av origin. The Danish HA sequences of H1av origin obtained in this study are marked in red, whereas the reference sequences of H1av are marked in black. The sequences belonging to the 10 recently defined European cluster (*Henritzi et al., 2020*) are highlighted with the name of the specific cluster in gray color. Two clusters are highlighted with brackets; one containing only Danish sequences and one containing H1av sequences derived from H1avNx viruses containing mainly and internal gene cassette of H1N1pdm09 origin. The reference sequences of IRD used to define the 1. C.1 (1C1) and 1.C.2 (1C2) clades are marked with a green *. Node labels represent posterior probabilities. All sequences are named according to the influenza nomenclature, and the accession number is given as a suffix for each sequence. A_Arnsberg_6554_1979_H1N1_KT715451 was used as an outgroup.

The online version of this article includes the following figure supplement(s) for figure 2:

**Figure supplement 1.** Strict molecular clock tree of the H1av nucleotide sequences of the Danish swine influenza A virus (swIAV) surveillance 2011–2018.

of the swine-derived H1pdm09 reference sequences, thereby containing only H1pdm09 sequences of swine origin. The cluster was therefore termed the swine-like cluster 'Sw-L cluster' (*Figure 3*: Sw-L cluster). The remaining 18 Danish sequences from this study were branching out basally to this cluster together with both human seasonal H1pdm09 and swine-derived H1pdm09 sequences. The 30 H1pdm09 sequences of the Sw-L cluster were collected between 2015 and 2018, whereas the 18 H1pdm09 sequences outside this cluster were collected between 2013 and 2017. The strict molecular clock tree and the TempEst analysis of the Danish H1pdm09 sequences suggested that the sequences evolved according to time with stable substitution rate of $4.9 \times 10^{-3}$ per site per year (*Figure 3—figure supplement 1* and *Table 1*) and that the common ancestor for the sequences of

**Table 1.** Results of the evolutionary analysis of the hemagglutinin (HA) protein of the H1av and the H1pdm09 lineages obtained in the Danish swine influenza A virus (swIAV) surveillance 2011–2018, including division of the H1pdm09 sequences into the ones located in and outside the Sw-L cluster.

The best-fitting substitution model M1a and M2a indicates neutral/negative selection and positive selection, respectively. The global ω ratio indicates average dN/dS ratio of the sequences. The positions in which positive selection occurred are given, and the positions are numbered from the first methionine. The TempEst correlation coefficient indicates how well the sequences are accumulating nucleotide changes proportionally to elapsed time. The substitution rate was calculated based on the length of the HA gene (1701nts) and represents the number of substitutions for the entire gene per year.

| | H1pdm<br>n = 48 | H1pdm09<br>Sw-L cluster<br>n = 30 | H1pdm09 outside the Sw-L cluster<br>n = 18 | H1av<br>n = 78 |
|---|---|---|---|---|
| Probability of M1a/ M2a (%) | 0.1/99.9 | 91/9 | 0.1/99.9 | 0.1/99.9 |
| Global ω ratio | 0.27 | 0.25 | 0.26 | 0.19 |
| Positions with positive selection (antigenic site/RBS) | 142K (Sa), 154P (Ca1/RBS), 172E (Sa), 174V (Sa), 200A (RBS), 202S, 204D (RBS), 206R (Sb/RBS), 207T (Sb/RBS) | - | 142K (Sa), 154P, 159K (Ca1/RBS), 160G, 172E (Sa), 178L (Sa), 179N (Sa), 200A (RBS), 202S, 203D (Sb/RBS), 204D (RBS), 206R (Sb/RBS), 207T (Sb/RBS), 338I, 391G | 142Q (Sa), 159K (Ca1/RBS), 172R (Sa), 173E (Sb) |
| TempEst correlation coefficient | 0.87 | 0.85 | 0.93 | 0.56 |
| Substitution rate | $4.9 \times 10^{-3}$ per site per year = 8.3 nt substitutions per year | $4.6 \times 10^{-3}$ per site per year = 7.8 nt substitutions per year | $6.1 \times 10^{-3}$ per site per year = 10.4 nt substitutions per year | $4.6 \times 10^{-3}$ per site per year = 7.8 nt substitutions per year |

the Sw-L cluster was dated around 2014. To reveal the genetic differences of the sequences of the Sw-L cluster and the sequences located outside of the cluster including the reference H1pdm09 sequences of human and swine origin, the nucleotide sequences were translated into amino acids and compared. This analysis revealed that 20 aa positions were important in defining the Sw-L cluster from the remaining sequences, 6 of those were a 100% unique for the sequence of the cluster. Five of these six positions were located either in previously defined antigenic sites (Ca and Sb) or the receptor binding site (RBS) (*Table 2*). To investigate if any aa residues were specific to all the swine-derived H1pdm09 sequences compared to all the human seasonal sequences, a similar comparison of amino acids was performed. This analysis could reveal mutations occurring due to adaptation to swine. However, no unique swine or human residues were revealed in this analysis. Nonetheless, at position 273, significantly more swine H1pdm09 proteins (91%) carried an A compared to the human seasonal H1pdm09 proteins (27%) ($p<0.05$).

The CODEML analysis for determining the best-fitting substitution model revealed that the M2a model fitted the Danish H1pdm09 sequences significantly better than the M1a model, providing strong evidence for positive selection occurring in the HA protein. Moreover, the dN/dS ratios for individual codons under the M2a model strongly suggested the presence of positive selection in nine aa positions all situated in the globular part of the HA protein and eight located specifically in antigenic sites or in the RBS (*Table 1*). The same analysis was repeated separately on sequences of the Sw-L cluster and the remaining Danish H1pdm09 sequences of the study. Interestingly, these analyses revealed that positive selection dominated the Danish H1pdm09 sequences located outside the Sw-L cluster as the M2a model fitted the sequences significantly better than the M1a model. Additionally, the dN/dS ratios for individual codons under the M2a model strongly suggested the presence of positive selection in 15 different aa positions, including 10 positions located in antigenic sites or the RBS. Conversely, model M1a fitted the Sw-L sequences significantly better, suggesting that no positive selection occurred among these sequences. The strict molecular clock and TempEst analysis were also repeated for the two groups of H1pdm09 sequences. Interestingly, the H1pdm09 sequences located outside the Sw-L cluster had a higher substitution rate and also showed a higher correlation coefficient in the TempEst analysis compared to the sequences of the Sw-L cluster (*Table 1*). In summary, different selection pressures were evident in the Danish H1pdm09 sequences

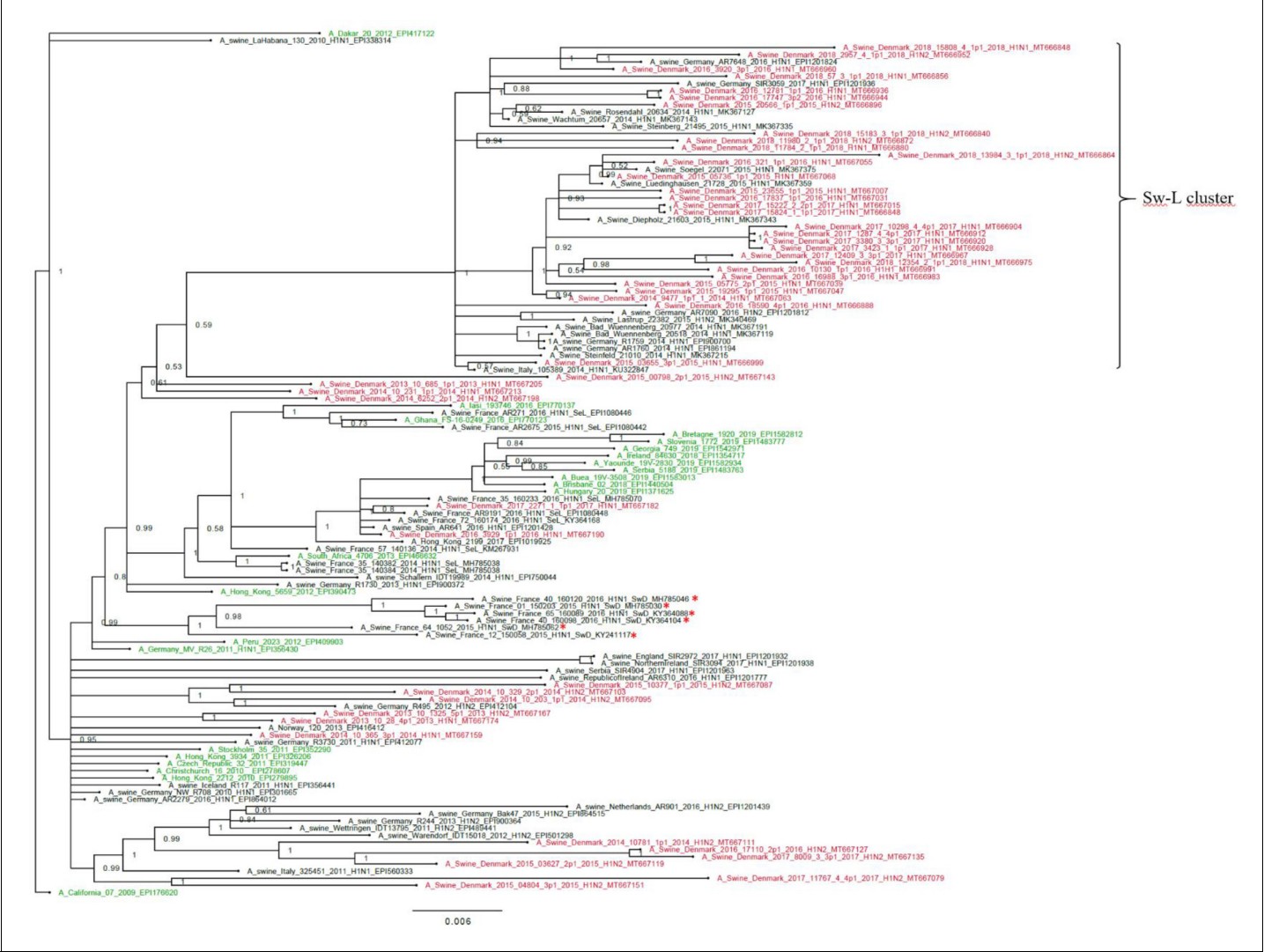

**Figure 3.** Bayesian phylogenetic tree of the Danish hemagglutinin (HA) nucleotide sequences of H1N1pdm09 origin from 2011 to 2018 and HA reference sequences of H1N1pdm09 origin. The Danish HA sequences of H1N1pdm09 origin obtained in this study are marked in red, whereas the official human reference sequences of H1N1pdm09 origin of each year from 2009 to 2019 are marked in green. The remaining sequences are HA reference sequences of H1N1pdm09 origin obtained from swine and include representative sequences of each of the five clusters defined in a recent paper by *Henritzi et al., 2020*, and the swine divergent (SwD and marked with a red *) and seasonal-like (SeL) of an earlier French study by *Chastagner et al., 2018*. Moreover, the closest swine HA sequence matches of NCBI GenBank were included. Sw-L cluster includes the cluster formed by only H1pdm09Nx sequences of swine origin. All sequences are named according to the influenza nomenclature, and the accession number is given as a suffix for each sequence. A_California_07_2009_EPI176620 was used as an outgroup. Node labels represent posterior probabilities.

The online version of this article includes the following figure supplement(s) for figure 3:

**Figure supplement 1.** Strict molecular clock tree of the H1pdm09 sequences of the Danish swine influenza A virus (swIAV) surveillance 2011–2018.

**Figure supplement 2.** Strict molecular clock tree of the N1pdm09 sequences of the Danish swine influenza A virus (swIAV) surveillance 2011–2018.

**Figure supplement 3.** Strict molecular clock tree of the N1av sequences of the Danish swine influenza A virus (swIAV) surveillance 2011–2018.

**Figure supplement 4.** Strict molecular clock tree of the N2dk sequences of the Danish swine influenza A virus (swIAV) surveillance 2011–2018.

**Figure supplement 5.** Strict molecular clock tree of the N2hu95 sequences of the Danish swine influenza A virus (swIAV) surveillance 2011–2018.

of the study, with a significant higher selection pressure being identified in the H1pdm09 sequences located outside the Sw-L cluster.

Additionally, differences in N-linked and O-linked glycosylation sites between the Danish H1pdm09 sequences of the Sw-L cluster and the H1pdm09 sequences located outside the cluster were examined. The results revealed that all proteins of both groups of sequences were predicted

**Table 2.** Mutations of the H1pdm09 defining the sequences of the Sw-L cluster of the H1pdm09 phylogenetic tree (**Figure 3**). Positions in the aa sequences are numbered from the first methionine. Bold letters indicate the mutations that are unique to the Danish swine divergent cluster. RBS: receptor binding site.

| Aa change Hu-L → Sw-L | Prevalence in the Sw-L cluster | Prevalence outside the Sw-L cluster | Antigenic site/RBS |
|---|---|---|---|
| N/S16D | 44/46 | 0/75 | - |
| N/D114N/H | 46/46 | 39/75 | - |
| **P141T** | 46/46 | 0/75 | Sa |
| N/G/K142D | 45/46 | 15/75 | Sa |
| **H143N/D/E** | 46/46 | 0/75 | Sa |
| D144N/K | 42/46 | 0/75 | - |
| S145L | 41/46 | 0/75 | - |
| **N/S/D146K/E** | 46/46 | 0/75 | RBS |
| N/Q/K147E | 46/46 | 1/75 | RBS |
| A152S | 46/46 | 2/75 | RBS |
| N/K/R159S | 45/46 | 1/75 | Ca1/RBS |
| **G/E/R/V172T/M** | 46/46 | 0/75 | Sa |
| N/S/K173D/G | 44/46 | 0/75 | Sb |
| **K/E/A/Q180I** | 46/46 | 0/75 | Sa |
| D185N | 39/46 | 0/75 | Ca1 |
| T/S/N/D202A | 44/46 | 0/75 | - |
| S/T/W/E207R | 42/46 | 3/75 | Sb/RBS |
| D239N | 46/46 | 4/75 | Ca2/RBS |
| **K/T/E319Q** | 46/46 | 0/75 | - |
| I/V/T338D | 42/46 | 0/75 | - |

to be N-glycosylated at position 28, 40, 304, and 557 (numbering from the first methionine). Significantly more sequences of the Sw-L cluster (80%) had an O-linked glycosylation site at position 150 compared to the sequences located outside the cluster (11%) (p<0.05). Interestingly, position 150 is located in the RBS. Conversely, significantly more sequences located outside the Sw-L cluster (44%) had a O-linked glycosylation site at position 145 compared to the sequences of the Sw-L cluster (3%) (p<0.05). Position 145 is also located in the vicinity of the RBS.

Previously defined residues of the HA proteins regarded as important for host-adaptation, pathogenicity, receptor binding, and virulence were examined and compared between H1avNx and H1pdm09Nx strains. The results are listed in *Supplementary file 2*.

The three H3 sequences obtained in this study showed a low nucleotide diversity (pi) of 0.027, SE 0.002. The closest human IAV match in NCBI GenBank for all of the three sequences was 'A/Denmark/129/2005(H3N2)' with accession number EU103786. As only three sequences were obtained, no further phylogenetic or evolutionary analysis was performed.

## Neuraminidase characteristics

In total, 32 N1pdm, 14 N1av, 75 N2dk, and 8 N2hu95 full-length NA sequences were obtained and analyzed separately according to the lineage.

The N1pdm09 nucleotide diversity (pi) was 0.029, SE 0.0005. The majority of the sequences obtained between 2015 and 2018 were located in one cluster, whereas the oldest sequences (2013–2014) were located outside the cluster (*Figure 3—figure supplement 2*). The TempEst analysis showed a high correlation coefficient similar to that of the H1pdm09 sequences, indicating that the genetic divergence evolved according to time. The Beast analysis revealed a substitution rate of $3.9 \times 10^{-3}$ per site per year. However, no evidence of positive selection was revealed (*Table 3*).

The N1av nucleotide diversity was 0.097, SE 0.003, similar to the nucleotide diversity of the H1av nucleotide sequences. No clear clustering was observed in the Bayesian tree (*Figure 3—figure*

**Table 3.** Results of the evolutionary analysis of the neuraminidase (NA) gene of the N1pdm, N1av, N2dk, and N2hu95 lineages.

The best-fitting substitution model M1a and M2a indicates neutral/negative selection and positive selection, respectively. The global ω ratio indicates average dN/dS ratio of the sequences. The positions in which positive selection occurred are given, and the positions are numbered from the first methionine. The TempEst correlation coefficient indicates how well the sequences are accumulating nucleotide changes proportionally to elapsed time. The substitution rate was calculated based on the length of the NA gene (1413nts) and represents the number of substitutions for the entire gene per year.

|  | N1pdm | N1av | N2dk | N2hu95 |
|---|---|---|---|---|
| Probability of M1a/M2a (%) | 90/10 | 73/27 | 88/12 | 88/12 |
| Global ω ratio | 0.24 | 0.15 | 0.17 | 0.18 |
| Positions with positive Hu-Lection | - | - | - | - |
| TempEst correlation coefficient | 0.88 | 0.68 | 0.75 | 0.62 |
| Substitution rate (per site per year) | $3.9 \times 10^{-3}$ | $5.9 \times 10^{-3}$ | $4.4 \times 10^{-3}$ | N.A |

supplement 3). The TempEst analysis showed a relative low correlation between the genetic divergence and time, and the Beast analysis revealed a substitution rate of $5.9 \times 10^{-3}$ per site per year. No evidence of positive selection was observed (*Table 3*).

The N2dk nucleotide diversity was 0.08, SE 0.0005, and the Bayesian analysis revealed six main clusters. Each cluster contained sequences dispersed over the majority of the surveillance period, suggesting no temporal clustering. Interestingly, one major cluster only contained HxN2dk from strains having a full or partial H1N1pdm09 internal gene cassette. Moreover, this cluster contained 28/30 of the same samples as cluster 3 of the H1av sequences, which also clustered according to the origin of the internal gene cassette (*Figure 2* and *Figure 3—figure supplement 4*). The TempEst and Beast analysis revealed a low correlation between genetic divergence and time, and a substitution rate of $4.4 \times 10^{-3}$ per site per year. As for the other NA lineages, no evidence of positive selection was observed (*Table 3*).

The eighth N2hu95 sequences showed a sequence diversity of 0.085, SE 0.006, and despite the limited number of sequences, the Bayesian phylogenetic analysis revealed two main clusters: one containing sequences derived from viruses containing a full avian internal gene cassette and one only containing sequences with a full or partial H1N1pdm09 internal cassette (*Figure 3—figure supplement 5*). The TempEst analysis revealed a low correlation between genetic divergence and time, and the low number of sequences resulted in an overestimated substitution rate, which therefore was not included in the results. No evidence of positive selection was observed (*Table 3*).

All of the NA sequences across the different lineages (n = 129) were examined for specific aa changes encoding either NA resistance or increased virulence. However, none of the NA sequences had any of these aa changes.

## The internal gene cassette

In total, 17 different genotypes were identified in this study (*Table 4*). The lineages H1N2dk, H1avN1av, and the H1avN2hu95 showed the highest number of diverse genotypes, whereas most strains including at least one surface gene of H1N1pdm09 origin had a complete internal gene cassette of H1N1pdm09 origin. Detailed information on the origin of all gene segments of all individual samples is listed in *Supplementary file 1*.

All internal gene full-length sequences (M, NP, NS, PA, PB1, and PB2) were subjected to individual Bayesian phylogenetic analysis, which for all gene segments revealed two main clusters: one containing sequences of H1avN1av origin and one of H1N1pdm09 origin (*Table 4—source data 1–6*).

Full-genome sequencing of the swIAV isolates obtained over the 8 years revealed that since 2013 an increasing number of the H1N2dk lineage sequenced had acquired an internal gene cassette of H1N1pdm09 origin (*Table 4—source data 7*). Similarly, though not as many samples were available, the H1avN2hu95 also seemed to gain internal genes of H1N1pdm09 origin over time. In contrast, the H1avN1av, roughly maintained an avian-like swine internal gene cassette, with an exception of

**Table 4.** Genotypes of the different Danish swine influenza A virus (swIAV) isolates from 2013 to 2018.

Previously described genotypes originated from *Watson et al., 2015* (*) and *Henritzi et al., 2020* (). The total number of samples obtained for each genotype is given in the column 'n', and the total number of samples does not correspond to the total number of samples sequences since sequences of each segment were not obtained from each sample; see *Supplementary file 1*.

| | HA | NA | M | NP | NS | PA | PB1 | PB2 | Genotype | n |
|---|---|---|---|---|---|---|---|---|---|---|
| **H1N2dk** | | | | | | | | | | |
| Genotype 1 | blue | blue | blue | blue | blue | blue | blue | blue | D* | 18 |
| Genotype 2 | blue | blue | blue | green | green | green | green | green | - | 1 |
| Genotype 3 | blue | blue | green | green | blue | green | green | green | AH˙ | 1 |
| Genotype 4 | blue | blue | green | green | green | green | green | green | T* | 28 |
| **H1avN1av** | | | | | | | | | | |
| Genotype 1 | blue | blue | blue | blue | blue | blue | blue | blue | A* | 9 |
| Genotype 2 | blue | blue | green | blue | blue | blue | blue | blue | M* | 1 |
| Genotype 3 | blue | blue | green | green | blue | blue | blue | blue | AB˙ | 1 |
| Genotype 4 | green | green | green | green | green | green | green | green | - | 1 |
| **H1N2hu95** | | | | | | | | | | |
| Genotype 1 | blue | red | blue | blue | blue | blue | blue | blue | I* | 2 |
| Genotype 2 | blue | red | green | blue | blue | blue | blue | blue | - | 1 |
| Genotype 3 | blue | red | blue | blue | blue | blue | blue | blue | - | 1 |
| **H1N1pdm09** | | | | | | | | | | |
| Genotype 1 | green | green | green | green | green | green | green | green | P* | 31 |
| **H1pdm09N2dk** | | | | | | | | | | |
| Genotype 1 | green | blue | green | green | green | green | green | green | - | 1 |
| Genotype 2 | green | blue | green | green | green | green | green | green | R* | 12 |
| **H1pdm09N2hu95** | | | | | | | | | | |
| Genotype 1 | green | red | green | green | green | green | green | green | - | 4 |
| **H3hu05N2dk** | | | | | | | | | | |
| Genotype 1 | red | blue | green | green | green | green | green | green | - | 3 |
| **H1avN1pdm09** | | | | | | | | | | |
| Genotype 1 | blue | green | green | green | green | green | green | green | - | 1 |

**Color-code:**

| Enzootic swine origin (H1avN1av, H1N2dk, H3N2sw) | Seasonal human H3N2 origin | H1N1pdm09 origin |
|---|---|---|

The online version of this article includes the following source data for Table 4:

**Source data 1.** Bayesian phylogenetic tree of the M sequences of the Danish swine influenza A virus (swIAV) surveillance 2011–2018.A/sw/Denmark/ 12687/2003(H1N2) accession number: KC900267 was used as the outgroup. A blue taxon indicates that the M gene of the sample is of avian-like origin, whereas a black taxon indicates that the M gene of the sample is of H1N1pmd09 origin. Sequences are named according to their sequence ID and lineage.

**Source data 2.** Bayesian phylogenetic tree of the NP sequences of the Danish swine influenza A virus (swIAV) surveillance 2011–2018.A/sw/Denmark/ 12687/2003(H1N2) accession number: KC900267 was used as the outgroup. A blue taxon indicates that the NP gene of the sample is of avian-like origin, whereas a black taxon indicates that the NP gene of the sample is of H1N1pmd09 origin. Sequences are named according to their sequence ID and lineage.

**Source data 3.** Bayesian phylogenetic tree of the NS sequences of the Danish swine influenza A virus (swIAV) surveillance 2011–2018.A/sw/Denmark/ 12687/2003(H1N2) accession number: KC900267 was used as an outgroup. A blue taxon indicates that the NS gene of the sample is of avian-like origin, whereas a black taxon indicates that the NS gene of the sample is of H1N1pmd09 origin. Sequences are named according to their sequence ID and lineage.

**Source data 4.** Bayesian phylogenetic tree of the PA sequences of the Danish swine influenza A virus (swIAV) surveillance 2011–2018.A/sw/Denmark/ 12687/2003(H1N2) accession number: KC900267 was used as an outgroup. A blue taxon indicates that the PA gene of the sample is of avian-like origin, whereas a black taxon indicates that the PA gene of the sample is of H1N1pmd09 origin.

**Source data 5.** Bayesian phylogenetic tree of the PB1 sequences of the Danish swine influenza A virus (swIAV) surveillance 2011–2018.A/sw/Denmark/ 12687/2003(H1N2) accession number: KC900267 was used as an outgroup. A blue taxon indicates that the PB1 gene of the sample is of avian-like origin, whereas a black taxon indicates that the PB1 gene of the sample is of H1N1pmd09 origin.

**Source data 6.** Bayesian phylogenetic tree of the PB2 sequences of the Danish swine influenza A virus (swIAV) surveillance 2011–2018.A/sw/Denmark/ 12687/2003(H1N2) accession number: KC900267 was used as an outgroup. A blue taxon indicates that the PB2 gene of the sample is of avian-like origin, whereas a black taxon indicates that the PB2 gene of the sample is of H1N1pmd09 origin.

**Source data 7.** Percentage of H1N2dk isolates containing at least one internal gene of H1N1pdm09 origin.

three isolates, which contained an NP gene, an M gene and the NS, NP, PA, PB1, and PB2 genes of H1N1pdm09 origin, respectively. All other reassortants including at least one surface gene of H1N1pdm09 origin (H1N1pdm09, H1pdmN2dk, H1avN1pdm09, and H1pdmN1av) contained a complete H1N1pdm09 internal gene cassette, with the exception of one H1pdmN2dk virus (A/sw/Denmark/2013-10-1325-5p1/2013(H1N2): MT667167), which had an M-gene of H1avN1av origin. In addition, all the three full-genome sequences of the H3hu05N2dk viruses contained a complete internal gene cassette of H1N1pdm09 origin (*Supplementary file 1*).

Previously defined important residues of the proteins encoded by the internal gene cassette were analyzed, and the results are summarized in *Supplementary file 2*. Furthermore, comparisons of the proteins encoded by the internal gene cassette of the sequences of the Sw-L cluster and the Danish H1pdm09 sequences located outside the Sw-L cluster were performed, and some aa differences between the two groups were identified. However, none of the aa differences were 100% specific to each group (*Supplementary file 3*).

## Discussion

### Seasonality

IAV infections in swine have been considered a disease of late autumn and early winter (*Diaz et al., 2015*; *Walia et al., 2019*; *Chamba Pardo et al., 2017*; *Kyriakis et al., 2011*), but the results reported here reveal no significant differences between the different seasons. This supports the recent studies describing the enzootic persistence of swIAV (*Ryt-Hansen et al., 2019a*; *Simon-Grifé et al., 2012*; *Rose et al., 2013*; *Ryt-Hansen et al., 2020*), most likely as a consequence of the herd sizes and management procedures under the current conditions of commercial swine herds. A similar lack of seasonality was found in other countries with comparable management structures (*Simon et al., 2014*; *Kyriakis et al., 2011*; *Harder et al., 2013*). Inadequate information on the severity of clinical signs was available for the Danish submissions, but a recent study from France revealed that the clinical symptoms encountered during the winter months were more severe (*Hervé et al., 2019*), which may explain the observed increase in the number of submissions in the autumn and winter. The increase in submissions during the autumn and winter may also be explained by the seasonal appearance of other respiratory pathogens such as mycoplasma and other bacteria (*Eze et al., 2015*). Finally, some veterinarians are still considering swIAV to be a seasonal disease and are therefore not submitting samples for swIAV testing during summer. No seasonality was documented for the prevalence of H1pdm09-positive submissions either, which is in accordance with a recent French study (*Chastagner et al., 2018*). This could indicate that while H1N1pdm09 reverse-zoonosis events occurs during the human influenza season, the high level of H1N1pdm09 circulating in Danish pigs independent of the human influenza season hides the impact observed on the H1N1pdm09 occurrence during the autumn and winter months.

### Prevalent lineages and reassortant swIAV

During the first three years of the surveillance program, the two most common IAV lineages in Danish swine were H1avN1av and H1N2dk, which harbor the same HA gene. However, soon after the first introduction of H1N1pdm09 in January 2010, this lineage rapidly spread and has almost replaced the H1avN1av. The H3N2 has almost disappeared from Denmark, in line with surveillance data obtained in some other European countries such as the United Kingdom and France (*Simon et al., 2014*; *Kyriakis et al., 2011*). Concurrently, the H1N2dk has increased in prevalence and has gradually gained an internal gene cassette of H1N1pdm09 origin, suggesting that this gene constellation is beneficial for the virus. In general, an increase in Danish swIAV carrying an internal gene cassette of H1N1pdm09 origin was observed, which indicates that an internal gene cassette of H1N1pdm09 origin is advantageous. The benefit of having a complete or partial internal gene cassette of H1N1pdm09 origin could be explained by the polymerase genes having a better/increased replication efficiency (*Ma et al., 2015*). In addition, certain gene combinations might enhance the transmissibility of the virus, for example, the M-gene of H1N1pdm09 origin in combination with the A/Puerto Rico/8/34 (H1N1) strain showed increased transmissibility in the guinea pig model (*Chou, 2011*). Interestingly, based on the phylogenetic trees, it seems that the internal gene cassette might have an influence on the evolution of the surface genes as several clusters among the H1av,

N2dk, and N2hu95 sequences correlate with the origins of the internal gene cassette. However, further studies are needed to investigate how the different gene segments can influence each other, but it might be related to the specific reassortment event forming a common ancestor for the cluster. Finally, the replacement of the avian internal gene cassette with an H1N1pdm09 internal gene cassette could enhance the zoonotic potential as proposed for the American H3N2v (*Bowman et al., 2012*) and the British H1N2hu (*Everett et al., 2020*) lineages, which have resulted in several human infections. Therefore, the pandemic potential of swIAV harboring gene segments of H1N1pdm09 origin should be a future research focus.

Several novel reassortants and genotypes were identified during the 8-year surveillance period. These findings underline the importance of having a national swIAV surveillance program, which acts as an early warning system both for the swine industry and for the human health sector, ensuring that novel reassortants and variants escaping current vaccines can be timely identified. The H3hu05N2dk reassortant is a perfect example hereof as it is a triple-reassortant swIAV including gene segments from IAV of enzootic swIAV origin, H1N1pdm09 origin, and human seasonal IAV origin (*Krog et al., 2017*). Surprisingly, this reassortant has only been sporadically detected during the last 5 years. A possible dissemination of this reassortant among Danish swine herds would probably have devastating consequences because there is no population immunity towards the human seasonal H3hu05 (*Krog et al., 2017*). This indicates that other factors than preexisting immunity towards the HA protein could be important for the spread of novel swIAV reassortants and strains. Indeed, the most successful virus in Denmark during the last 7 years has been the H1N2dk, despite there being a high level of population immunity towards the HA protein of this lineage since the 1990s. Combined with the findings that the internal cassette of H1N1pdm09 origin seems to benefit viral competitiveness, we might need to change our perception that preexisting immunity to HA is the only driver of IAV evolution. Antigenic testing of different H1N2dk would aid in elucidating this hypothesis, but this was beyond the scope of this study. Another case of spillover of human seasonal IAV was observed in the reassortants H1avN2hu95 and H1pdmN2hu95, both containing an NA gene derived from the human flu of 1995 (*Breum et al., 2013*). The circulation of these two reassortants is worrying from a zoonotic perspective because seven and eight gene segments of these two viruses, respectively, originate from human IAVs, and therefore must be somewhat adapted to replicate in- and spread between humans. Especially, the H1avN2hu95 reassortant needs attention as spillover into the human population would introduce a novel HA gene, to which the level of cross-immunity is limited (*Sun et al., 2020*; *Henritzi et al., 2020*).

## Genetic and antigenic drift

Another important aspect of swIAV evolution is the genetic drift, mainly affecting the two surface genes (HA and NA) (*Kim et al., 2018*; *Carrat and Flahault, 2007*). Especially, the H1av seem to have undergone extensive genetic and antigenic drift as a great sequence diversity was revealed. It was evident that the evolution of the H1av gene did not evolve in one specific direction over time, but rather evolved in many different directions, resulting in a vast number of different H1av clusters and variants. This confirms the results of a recent study investigating the genetic and antigenic diversity of the H1avNx lineage (*Henritzi et al., 2020*). In a previous study (*Ryt-Hansen et al., 2020*), we found that the evolution of the H1av in a single herd followed a pectinate pattern mirroring the pattern seen globally for the human seasonal influenza strains, which contradict the general perception that swine IAV is not prone to selection driven by preexisting immunity like in humans. In the present study, we assessed the H1av evolution over time in the Danish pig population as a whole and over several years and failed to confirm this pectinate pattern. We speculate that swIAV evolution at the single herd level is identical to the pattern seen in the global human population, but when the swIAV evolution is evaluated on a national or global scale, this pectinate pattern is disrupted as observed in this study and others (*Henritzi et al., 2020*). The reason for this difference is probably that the human population, due to the extensive global interactions, can be regarded as a single 'epidemiological unit,' whereas swine herds, due to a high level of external biosecurity and limited exchange of live animals between herds, represent a vast variety of closed 'epidemiological units,' which each have a specific and probably pectinate pattern of evolution. Still, despite the lack of a clear pectinate like evolution, the analysis showed that H1av protein had clearly undergone positive selection on specific codons located in antigenic sites, which is known to alter the binding of neutralizing antibodies (*Ryt-Hansen et al., 2019b*; *Matsuzaki et al., 2014*; *Rudneva et al., 2012*). This further confirms

our previous findings that the herd immunity leads to significant antigenic drift in the globular head of the HA protein as seen for human seasonal IAV (*Petrova and Russell, 2018*). Moreover, the fact that similar residues were targets for positive selection between different herds could indicate that some residues in the HA protein are of particular importance for swIAV evolution. Finally, the substitution rate estimated for H1av was similar to that documented in previous studies (*Furuse et al., 2010*; *Moreno et al., 2013*; *Marozin et al., 2002*; *Lam et al., 2008*), but was lower than the substitution rate estimated for H1av in a single herd over time (*Ryt-Hansen et al., 2020*). This emphasizes that one should differentiate when comparing evolutionary results based on data obtained from a single herd or data obtained through extensive surveillance programs.

The H1pdm09 sequences analyzed in this study revealed the existence of a cluster only containing sequences derived from swine (Sw-L cluster). The H1pdm09 sequences located outside the cluster were scattered among human seasonal H1pdm09 sequences. The same pattern has been described in two other European studies (*Chastagner et al., 2018*; *Henritzi et al., 2020*), and similarly we speculate that these two groups of sequences represent a swine-adapted group of H1pdm09NX viruses and a group of viruses derived from reverse-zoonotic events involving human seasonal flu. Comparison of the aa sequences between Sw-L cluster and the HA proteins of viruses located outside this cluster revealed 20 aa differences including 6 aa residues unique to the Sw-L cluster. The majority of the aa differences were located in previously defined antigenic sites and the RBS, indicating that these mutations could have a probable relation to host adaption.

Interestingly, the sequences of the Sw-L cluster in this study clustered separately from the 'swine-derived' cluster documented in France (*Chastagner et al., 2018*), indicating the evolution of swIAV follows different evolutionary traits in populations that are not epidemiologically connected. The antigenic cartography performed on H1N1pdm09 viruses collected in France showed a high degree of cross-protection between the swine-adapted and the human seasonal-like H1N1pdm09 viruses isolated in 2014–2016 (*Chastagner et al., 2018*). Nonetheless, it should be taken into consideration that the Sw-L cluster of our study showed several changes different from the French swine divergent cluster, and therefore the antigen cartography should be repeated on the Danish H1pdmNx viruses of the Sw-L cluster. The presented data strongly indicate that the human seasonal H1N1pdm09 viruses still are capable of infecting swine, despite more than 10 years of adaption to humans, but it is unclear if the swine-adapted viruses of the Sw-L cluster also have retained their capability to infect humans. Studies to investigate this in the ferret model are ongoing. From a zoonotic point of view, it is worrying that the H1N1pdm09 viruses seem to evolve in different directions in pigs and humans, especially if the swine-adapted Sw-L viruses retain their capacity to infect humans.

## Specific host and virulence markers

The HA proteins of the H1pdm09 viruses seem to be better adapted to elicit a strong receptor binding to the α2.6-linked sialic acid receptor compared to the H1av HA proteins. This may reflect that the H1pdm09 HA are descendants of the H1N1 'Spanish flu' strain (*Smith et al., 2009*) and by that have circulated in mammals for at least 100 years, whereas the H1av HA protein was first detected in a mammal (pig) in the 1980s (*Scholtissek et al., 1983*). In turn, these results could also explain why very few cases of zoonotic infection involving H1av have been registered (*Sun et al., 2020*; *Jong, 1988*; *Myers et al., 2007*). The fact that more of the H1pdm09Nx viruses located outside the Sw-L cluster had 'D' at position 225 in the HA protein supports the assumption that these H1pdmNx viruses are indeed more similar to human seasonal-like H1Npdm09 viruses compared to the viruses of the Sw-L cluster and also indicates that the G225D transition may be more important in humans than in swine. In addition, the eight aa residues defined to differ between avian IAVs and H1N1pdm09 origin viruses in the NP, PB1, PB2, and PA gene segments (*Chen and Shih, 2009*) were consistent with the residues observed in the two clusters (H1avN1av and H1N1pdm09) of the gene segments of this study. This suggests that these residues are indeed specific for H1pdmNx swIAV.

The recently identified residues 48Q, 98K, and 99K of the H1avN1av NP protein conferring MxA resistance (*Dornfeld et al., 2019*) were documented in the majority (81%) of the Danish NP protein of H1avN1av origin. MxA resistance is believed to play a role in the zoonotic and pandemic potential of avian and swine IAV (*Dornfeld et al., 2019*; *Mänz et al., 2013*), and there is therefore a potential increased risk of zoonotic transmission in the Danish herds, where circulation of swIAV strains carrying these three mutations is present.

As for the aa differences observed between the internal gene cassette of the Sw-L cluster and the sequences located outside the cluster, the T76A mutation in the PB2 protein has been linked to an elevated interferon response (*Du et al., 2018*). Furthermore, the M283I aa change in the PB2 protein has previously been linked to decreased virulence of avian H5 IAV (*Chen et al., 2020*; *Wang et al., 2017*) and the N456S aa change has, on the other hand, been linked to human adaptations of the H3N2 lineage (*Wen et al., 2018*). For the PB1 protein, the M317I aa change has been identified in a H2N2 after multiple passaging in chicken eggs to create a temperature-sensitive IAV strain (*Martínez-Sobrido et al., 2018*). Finally, the C241Y in the PA protein has been linked to mammalian adaptions of avian H5N1 viruses (*Yamaji et al., 2015*). In summary, several of the aa differences of the Sw-L cluster and the sequences located outside the cluster have previously been described to have an influence on the virulence, replication efficiency, or host-response/adaptation, thereby emphasizing that these changes could be important in the adaption of H1pdm09Nx viruses to swine. However, this needs to be investigated further.

### Importance of swIAV surveillance programs

The results generated in the passive surveillance program of swIAV performed in Denmark from 2011 to 2018 highlight the importance of such a program. The surveillance was essential in identifying novel reassortants and genotypes that circulate among Danish swine, and the knowledge supports veterinarians and farmers daily in selecting the most compatible swIAV vaccine and understanding the swIAV transmission dynamics in the herd. Moreover, novel reassortants not covered by the currently available vaccines were identified, thereby avoiding unnecessary use of vaccines and encouraging medical companies to prioritize vaccine updates. These vaccine updates are not only encouraged by identifying novel reassortants, but also by documenting the level of antigenic drift, which previously has been shown to affect the level of cross-protection between strains of the same lineage (*Ryt-Hansen et al., 2019b*). Concurrently, an antigenic map of the European H1av strains has revealed huge antigenic distances of H1avNx strains (*Henritzi et al., 2020*) emphasizing a lack of cross-protection even within the same lineage. The number of submissions for swIAV diagnostics increased in the last years of the surveillance, indicating that the program is useful for farmers and veterinarians. Moreover, an increase in the number of submissions positive for swIAV may indicate that swIAV infections represent an increasing problem in Danish swine herds or that there is increased focus on swIAV as an important pathogen in the herds. Finally, the ability of the program in identifying novel reassortants and strains that have an increased risk of being zoonotic is vital from a human health perspective as it can function as an early warning system for future human IAV pandemics.

## Materials and methods

### Samples

Samples, including lung tissues, nasal swabs and oral fluids, originating from swine herds experiencing clinical signs of acute respiratory disease, were submitted for routine diagnostic examinations at the Danish National Veterinary Institute by veterinary practitioners from 2011 until 2018. The submissions included 1–5 samples (yearly average: 2–2.9) from each herd.

### RNA isolation

Total RNA was extracted from lung tissue, nasal swab samples, or cell cultured virus isolates by RNeasy Mini Kit (QIAGEN, Denmark) automated on the QIAcube (QIAGEN) according to the instructions from the supplier. The samples were prepared for extraction as follows: 200 µl nasal swab sample or virus isolate were mixed with 400 µl RLT-buffer containing β–mercaptoethanol, whereas 30 mg of lung tissue was homogenized in 600 µl RLT-buffer containing β–mercaptoethanol for 30 s at 15 Hz using the TissueLyser II (QIAGEN).

Oral fluid samples were prepared by homogenization of 200 µl sample (30 s at 15 Hz) in a TissueLyser II (QIAGEN) followed by centrifugation (2 min at 10,000 rpm). Total RNA was extracted from 140 µl of the prepared oral fluid sample using the QIAamp Viral RNA Mini Kit (QIAGEN) automated on the QIAcube (QIAGEN) according to the instructions from the supplier.

The total RNA from all sample types was eluted in 60 µl RNase-free water and stored at −80℃ until further analysis. Positive and negative controls were included in all extractions.

## Detection of swIAV

The presence of swIAV was detected by an in-house modified version of a real-time RT-PCR assay detecting the M gene (De Vleeschauwer et al., 2009). The assay was performed in a total reaction volume of 25 µl using the RNA Ultrasense One-Step Quantitative RT-PCR System (Invitrogen), 3 µl of extracted RNA, 300 nM forward primer (RimF), 600 nM 5′-labeled reverse primer (MaR-FAM), 400 nM 3′-labeled probe (MaProbe). Details of the primers and probes are listed in *Supplementary file 4*. All reactions were analyzed on the Rotor-GeneQ machine (QIAGEN) using the following PCR conditions: 50℃ 30 min; 95℃ 2 min; 45 cycles of 95℃ 15 s, 55℃ 15 s (acquiring using 470 nm as source and 660 nm as detector), 72℃ 20 s; 95℃ 15 s; melt curve analysis by ramping from 50℃ to 99℃, wait for 90 s on pre-melt conditioning at first step, rising by 1℃ each step and wait for 5 s before acquiring. A positive and negative control were included in all runs.

## Test for the HA gene of H1N1pdm09 origin by specific real-time PCR

All swIAV-positive samples were tested for the presence of the HA gene of H1N1pdm09 origin (H1pdm09) by an in-house real-time RT-PCR assay detecting specifically the HA gene of the pandemic virus (*Supplementary file 4*). All reactions were analyzed in a Rotor-GeneQ machine (QIAGEN) using the following PCR conditions: 45℃ for 10 min; 95℃ for 10 min; 45 cycles of 95℃ for 15 s; 55℃ for 20 s; 72℃ for 30 s. In 2018, an additional assay targeting the H1pdm09 was implemented to increase the sensitivity of the H1pdm09 screening (*Supplementary file 4*). The two H1pdm09 assays were run as a multiplex on the Rotor-GeneQ machine (QIAGEN) using the following PCR conditions: 45℃, 20 min; 95℃, 15 min; 45 cycles: 94℃, 30 s; 55℃, 20 s; 60℃, 20 s. A positive and negative control were included in all runs.

## Lineage determination

From 2011 to 2014, the lineage of the swIAV-positive samples were determined using Sanger sequencing of the HA and NA genes according to a previously described PCR protocol (*Ryt-Hansen et al., 2019c*). The PCR products were purified using the High Pure PCR product Purification Kit (Roche, Denmark). Subsequently, the purified PCR products were sent for sequencing at LGC Genomics (Berlin, Germany) with primers comprised the 'pQE' part of the PCR primers (*Supplementary file 4*).

From 2015 to 2017, the lineage of the swIAV-positive samples was determined using a multiplex real-time RT-PCR assay strategy. Two multiplex reactions including primers and probes for H3hu05, N1pdm09, H1av, N2hu95 or H3, H1pdm09, N1av, N2dk, and N2hu95, respectively (*Ryt-Hansen et al., 2019a*; *Supplementary file 4*), were analyzed on the Rotor-GeneQ machine (QIAGEN) using the following PCR conditions: 50℃ for 20 min; 95℃ for 15 min; 40 cycles of 94℃ for 60 s, 60℃ for 90 s. In 2018, the lineage determination of the swIAV-positive samples was additionally performed on the Fluidigm PCR platform (AH Diagnostics, USA) according to a previously published protocol (*Goecke et al., 2018*). All runs on the Rotor-GeneQ and the Fluidigm included positive controls representing all the possible lineages targeted by the different assays along with a negative control.

## Virus isolation

Virus was isolated from selected swIAV-positive clinical specimens by inoculation of Madin Darby Canine Kidney (MDCK) cells following standard cell culture procedures. In short, 150 mg lung tissue was homogenized in 1.5 ml MEM (Invitrogen Carlsbad, CA) containing 1000 units/ml penicillin and 1 mg/ml streptomycin. Sterile filtrated inoculums were prepared in viral growth medium (MEM 1×, L-glutamin 2 mM, non-essential amino acids 1×, 100 units/ml penicillin, 100 µg/ml streptomycin, and TPCK-treated trypsin 2 µg/ml) using either 10% lung tissue homogenate or 20% nasal swab or oral fluid sample. The inoculum was added to 70% confluent MDCK cells for 45 min at 37℃ and 5% $CO_2$ followed by the addition of fresh viral growth medium after wash of the inoculated cells. After 3 days, the cell culture supernatant was harvested and tested for IAV by real-time RT-PCR.

## Full-genome sequencing

From 2013 to 2017, full-genome sequencing was performed on cell culture-propagated influenza virus samples, which had been subjected to full-length PCR amplification of all eight gene segments with in-house designed primers (*Supplementary file 4*) using SuperScript III OneStep RT-PCR System with Platinum Taq High Fidelity. The PCR conditions were as follows for each gene segment: HA: 55°C, 30 min, 94°C, 2 min, 4× (94°C, 30 s – 55°C, 30 s – 68°C, 180 s), 41× (94°C, 30 s – 68°C, 210 s) and 68°C, 10 min. NA: 54°C, 30 min, 94°C, 2 min, 4× (94°C, 30 s – 58°C, 30 s – 68°C, 180 s), 41× (94°C, 30 s – 68°C, 210 s) and 68°C, 10 min. M: 50°C, 30 min, 94°C, 2 min, 41× (94°C, 30 s – 56°C, 30 s – 68°C, 90 s) and 68°C, 10 min. Nucleoprotein (NP): 58°C, 30 min, 94°C, 2 min, 4× (94°C, 30 s – 54°C, 30 s – 68°C, 180 s) and 41× (94°C, 30 s – 68°C, 210 s) and 68°C, 10 min. Nonstructural protein (NS): 58°C, 30 min, 94°C, 2 min, 41× (94°C, 30 s – 55°C, 30 s, 68°C, 90 s) and 68°C, 10 min. Polymerase basic protein 1 (PB1) and polymerase acidic protein (PA): 52°C, 30 min, 94°C, 2 min, 4× (94°C, 30 s – 52°C, 30 s – 68°C, 180 s), 41× (94°C, 30 s – 68°C, 210 s) and 68°C, 10 min. Polymerase basic protein 2 (PB2): 55°C, 30 min, 94°C, 2 min, 4× (94°C, 30 s – 52°C, 30 s – 68°C, 180 s), 41× (94°C, 30 s – 68°C, 210 s) and 68°C, 10 min. Purified PCR products for all gene segments were pooled in equimolar quantity to a final amount of 1 µg and used for next-generation sequencing (NGS) on the Ion Torrent PGM sequencer. The NGS, including library preparation, was carried out at the Multi-Assay Core facility located at the Technical University of Denmark. In 2018, full-genome sequencing was performed on cell culture propagated virus samples using universal influenza primers (*Kai Lee, 2016*; *Supplementary file 4*). Library preparation and NGS on the Illumina MiSeq platform were conducted at the Statens Serum Institut, Denmark.

## Sequence analysis

Data obtained from Sanger sequencing and subsequent analyses of the consensus sequences were performed using CLC Main Workbench version 7.6.2–20.0.3 (CLC bio A/S, Aarhus, Denmark). Alignments of each gene segment were created using the MUSCLE algorithm (*Edgar, 2004*). Phylogenetic trees were constructed using a distance-based method with the neighbor joining algorithm and bootstrap analysis with 1000 replicates. The results were verified by using Maximum Likelihood Phylogeny. Sequences obtained by NGS were assembled using the features 'de novo assembly' and 'map read to references' using 22 reference sequences representing the different lineages of each gene segment in CLC Genomics Workbench 4.6.1–8.0.2 (CLC bio A/S). The subtype and lineage of each sample and gene segment were determined based on MUSCLE alignments, subsequent neighbor joining phylogenetic trees, and the function 'BLAST against NCBI.' Moreover, sequence alignments of each lineage of the two surface gene segments (H1pdm09, H1av, N1pdm09, N1av, N2hu95, and N2dk) were analyzed for the average nucleotide diversity (pi) using author's own software. For more detailed phylogenetic analysis, Bayesian trees of each gene segment (internal genes) and lineage (H1pdm09, H1av, N1pdm09, N1av, N2hu95, and N2dk) were constructed using the program MrBayes with the following settings; nst=mixed and rates=invgamma. The trees were run for 10,000,000 generations and a sample frequency of 500 (*Ronquist and Huelsenbeck, 2003*). Additional alignment and Bayesian tree was constructed for the H1av and the H1pdm09 genes including reference sequences. The reference sequences used for the H1av phylogenetic analysis included the global reference sequences used for the 'Swine H1 Clade Classification Tool' provided by IRD on http://www.fludb.org plus the closest sequence matches of the H1av sequences obtained in this study from NCBI GenBank. Moreover sequences of each of the 10 European H1av clusters defined by a recent paper (*Henritzi et al., 2020*) were included. For the H1pdm09 phylogenetic analysis, the official human seasonal H1pdm09 reference sequences from the years 2009–2019 were included along with the closest sequence matches of the Danish sequences of the Sw-L cluster from NCBI GenBank swine. Moreover, the H1pdm09 sequences from each of the five H1pdm09 European clusters described in a recent study (*Henritzi et al., 2020*) were included. The GISAID and NCBI GenBank accession numbers of the sequences are included in the sequence taxon of the trees. Convergence of the Bayesian analysis was checked using Tracer version 1.7.1 (*Rambaut et al., 2018*), and the results visualized in Figtree version 1.4.4 (*Rambaut, 2006*).

In addition to the Bayesian phylogenetic analyses, strict molecular clock trees were constructed for the surface gene segments of the lineages – H1pdm09, H1av, N1pdm09, N1av, N2hu95, and N2dk – to determine the temporal evolution and the substitution rate. However, before the trees

were constructed, all sequences were investigated for the presence of a temporal signal (i.e., whether nucleotide changes accumulate roughly proportionally to elapsed time) using the program TempEst (*Rambaut et al., 2016*) and evaluating the correlation coefficient. Subsequently, the alignments of each lineage were analyzed in the program BEAST2 version 2.5.2, where the model settings were as previously described (*Ryt-Hansen et al., 2019b*). Briefly, the HKY substitution model with gamma-distributed rates over sites was chosen along with a strict clock model including tip dates. The outcome of the analysis was visualized in Figtree version 1.4.4 (*Rambaut, 2006*) and convergence checked in Tracer version 1.7.1 (*Rambaut et al., 2018*).

The surface gene segments of the different lineages – H1pdm09, H1av, N1pdm09, N1av, N2hu95, and N2dk – were investigated for the presence of positive selection using the CODEML program of the PAML package as previously described (*Ryt-Hansen et al., 2019b*). Briefly, this was done by comparing the fits of CODEML's substitution models M1a and M2a (NSsites = 1 and 2). M1a includes two categories of codons – some under negative selection (dN/dS ratio < 1) and some codons where mutations are neutral (dN/dS ratio = 1). Model M2a includes three categories of codons – the same two as M1a plus an additional category of codons under positive selection (dN/dS ratio > 1). If M2a fits a dataset significantly better than M1a, then there is evidence of positive selection in some codons (and the identity of these codons is also determined during model fitting). The average dN/dS ratio (global $\omega$ ratio) of the surface gene segments of the different lineages – H1pdm09, H1av, N1pdm09, N1av, N2hu95, and N2dk – was also estimated using CODEML with the setting NSsites = 0.

All nucleotide sequences of each gene segment were translated into amino acid (aa), and MUS-CLE (*Edgar, 2004*) alignments were created using CLC Main Workbench 20.0.3 (CLC bio A/S). Subsequently, the alignments were manually examined to determine the presence of previously described aa differences and residues. Specifically, for the HA proteins these included residues unique to the H1pdmN2dk reassortant (*Starick et al., 2012*) and residues linked to receptor binding (*Matrosovich et al., 2000*), (*de Vries et al., 2011*). Moreover, the five previously defined antigenic sites Sa, Sb, Ca1, Ca2, and Cb of the H1 subtype (*Matsuzaki et al., 2014*; *Caton et al., 1982*; *Manicassamy et al., 2010*) and the RBS (*Sriwilaijaroen and Suzuki, 2012*) were annotated to the H1av and H1pdm09 proteins and investigated for divergence and correlations to codons with increased dN/dS ratios. For the NA protein, residues encoding NA inhibitor resistance were investigated (*Abed et al., 2006*). All PB2 proteins were examined for specific residues encoding virulence (*Massin et al., 2001*), pathogenicity (*Liu et al., 2018*), and host adaptation (*Taft et al., 2015*). The eight residues of the NP, PB1, PB2, and PA proteins proposed to differ between avian viruses and viruses of the H1N1pdm09 lineage (*Chen and Shih, 2009*) were also investigated. Finally, the three residues of the NP protein recently found to confer MxA resistance (*Dornfeld et al., 2019*) were examined. The two groups of the H1pdm09 proteins were examined for differences in the number and location of N-linked and O-linked glycosylation sites using the NetNGlyc 1.0 (*Bioinformatics, D, 2017a*) and NetOGlyc 4.0 (*Bioinformatics, D, 2017b*) servers from DTU Bioinformatics, Denmark.

## Statistics

Results of the screening for swIAV and H1pdm09 in each submission were analyzed in Microsoft Excel 2016 version 16.0.4993.1001 and GraphPad (*GraphPad software, 2021*). The proportions of swIAV-positive, swIAV-negative, and the proportion of H1pdm09-positive submissions compared to total number of swIAV-positive submissions were calculated for each month. The monthly average percentage of swIAV-positive submissions and the proportion of H1pdm09-positive submissions were calculated based on the results obtained from each month during the 8 years, and the differences in the pairwise percentages and proportions of swIAV and H1pdm09 submission were investigated using a Student's t-test and a Fisher's exact test in GraphPad (*GraphPad software, 2021*). To determine differences between the prevalence of a specific aa residue at a given position, a chi-squared test in GraphPad (*GraphPad software, 2021*) was utilized. p-values below 0.05 were considered statistically significant.

## Acknowledgements

We would like to acknowledge all the Danish herds that submitted samples for the surveillance. Moreover, we acknowledge the authors, originating and submitting laboratories of the sequences

that we obtained from GISAID's EpiFlu Database (http://www.gisaid.org) and NCBI GenBank (http://www.ncbi.nlm.nih.gov). The farmers or the medical company IDT Biologika GmbH paid the initial screening for the presence of swIAV in a submission, while the Danish Veterinary and Food Administration paid the remaining analyses. In addition, the work presented in this article was supported by Novo Nordisk Foundation (FluZooMark – grant # NNF19OC0056326)

## Additional information

### Funding

| Funder | Grant reference number | Author |
|---|---|---|
| Novo Nordisk Foundation | NNF19OC0056326 | Lars Erik Larsen |
| IDT Biologika GmbH | SwIAV surveillance | Pia Ryt-Hansen<br>Jesper Schak Krog<br>Solvej Østergaard Breum<br>Charlotte Kristiane Hjulsager<br>Anders Gorm Pedersen<br>Ramona Trebbien<br>Lars Erik Larsen |
| Ministeriet for Fø devarer, Landbrug og Fiskeri | SwIAV surveillance | Pia Ryt-Hansen<br>Jesper Schak Krog<br>Solvej Østergaard Breum<br>Charlotte Kristiane Hjulsager<br>Anders Gorm Pedersen<br>Ramona Trebbien<br>Lars Erik Larsen |

The funders had no role in study design, data collection and interpretation, or the decision to submit the work for publication.

### Author contributions

Pia Ryt-Hansen, Conceptualization, Data curation, Software, Formal analysis, Investigation, Visualization, Methodology, Writing - original draft, Project administration; Jesper Schak Krog, Formal analysis, Investigation, Visualization, Methodology, Writing - review and editing; Solvej Østergaard Breum, Formal analysis, Investigation, Methodology, Writing - review and editing; Charlotte Kristiane Hjulsager, Conceptualization, Formal analysis, Investigation, Methodology, Project administration, Writing - review and editing; Anders Gorm Pedersen, Software, Formal analysis, Supervision, Writing - review and editing; Ramona Trebbien, Conceptualization, Data curation, Formal analysis, Investigation, Methodology, Writing - review and editing; Lars Erik Larsen, Conceptualization, Data curation, Formal analysis, Supervision, Funding acquisition, Validation, Investigation, Methodology, Project administration, Writing - review and editing

### Author ORCIDs

Pia Ryt-Hansen (ID) https://orcid.org/0000-0003-4819-6869
Charlotte Kristiane Hjulsager (ID) http://orcid.org/0000-0001-7557-8876

### Decision letter and Author response

Decision letter https://doi.org/10.7554/eLife.60940.sa1
Author response https://doi.org/10.7554/eLife.60940.sa2

## Additional files

### Supplementary files

• Supplementary file 1. The genotype of all full-genome sequenced samples during the Danish swine influenza A virus (swIAV) surveillance 2011–2018.

• Supplementary file 2. Residues examined for specific mutations involving host adaptation, virulence, pathogenicity, and dominating residues differing between H1av and H1N1pdm09 origin viruses. Aa positions are numbered according to the first methionine if nothing else is indicated.

• Supplementary file 3. Amino acid differences in the internal proteins of the Danish sequences of the Sw-L cluster and the sequences located outside the cluster. Aa positions are numbered according to the first methionine.

• Supplementary file 4. Primers and probes used for detection, subtyping, and full-genome sequencing of swine influenza A virus (swIAV). Nucleotides are named according to the IUPAC codes. Purple color indicates the pQE part of the primers.

• Transparent reporting form

## Data availability

All sequences generated in the study have been uploaded in NCBI GenBank with accession numbers MT666225 - MT667233. All analyses of the sequences are included in the manuscripts or in the supplement files and figures.

The following datasets were generated:

| Author(s) | Year | Dataset title | Dataset URL | Database and Identifier |
|---|---|---|---|---|
| Ryt-Hansen P, Krog JS, Breum SO, Hjulsager CK, Pedersen AG, Trebbien R, Larsen LE | 2021 | Influenza A virus (A/swine/Denmark/2015_09973_1p1/2015 (H1N1)) segment 1 polymerase PB2 (PB2) gene, complete cds | https://www.ncbi.nlm.nih.gov/nuccore/MT666225 | NCBI GenBank, MT666225 |
| Ryt-Hansen P, Krog JS, Breum SO, Hjulsager CK, Pedersen AG, Trebbien R, Larsen LE | 2021 | Influenza A virus (A/swine/Denmark/2015_09973_1p1/2015 (H1N1)) segment 2 polymerase PB1 (PB1) and PB1-F2 protein (PB1-F2) genes, complete cds | https://www.ncbi.nlm.nih.gov/nuccore/MT666226 | NCBI GenBank, MT666226 |
| Ryt-Hansen P, Krog JS, Breum SO, Hjulsager CK, Pedersen AG, Trebbien R, Larsen LE | 2021 | Influenza A virus (A/swine/Denmark/2015_09973_1p1/2015 (H1N1)) segment 3 polymerase PA (PA) and PA-X protein (PA-X) genes, complete cds | https://www.ncbi.nlm.nih.gov/nuccore/MT666227 | NCBI GenBank, MT666227 |
| Ryt-Hansen P, Krog JS, Breum SO, Hjulsager CK, Pedersen AG, Trebbien R, Larsen LE | 2021 | Influenza A virus (A/swine/Denmark/2015_09973_1p1/2015 (H1N1)) segment 4 hemagglutinin (HA) gene, complete cds | https://www.ncbi.nlm.nih.gov/nuccore/MT666228 | NCBI GenBank, MT666228 |
| Ryt-Hansen P, Krog JS, Breum SO, Hjulsager CK, Pedersen AG, Trebbien R, Larsen LE | 2021 | Influenza A virus (A/swine/Denmark/2015_09973_1p1/2015 (H1N1)) segment 5 nucleocapsid protein (NP) gene, complete cds | https://www.ncbi.nlm.nih.gov/nuccore/MT666229 | NCBI GenBank, MT666229 |
| Ryt-Hansen P, Krog JS, Breum SO, Hjulsager CK, Pedersen AG, Trebbien R, Larsen LE | 2021 | Influenza A virus (A/swine/Denmark/2015_09973_1p1/2015 (H1N1)) segment 6 neuraminidase (NA) gene, complete cds | https://www.ncbi.nlm.nih.gov/nuccore/MT666230 | NCBI GenBank, MT666230 |
| Ryt-Hansen P, Krog JS, Breum SO, Hjulsager CK, Pedersen AG, Trebbien R, Larsen LE | 2021 | Influenza A virus (A/swine/Denmark/2015_09973_1p1/2015 (H1N1)) segment 7 matrix protein 2 (M2) gene, partial cds; and matrix protein 1 (M1) gene, complete cds | https://www.ncbi.nlm.nih.gov/nuccore/MT666231 | NCBI GenBank, MT666231 |

| Ryt-Hansen P, Krog JS, Breum SO, Hjulsager CK, Pedersen AG, Trebbien R, Larsen LE | 2021 | Influenza A virus (A/swine/Denmark/2015_09973_1p1/2015 (H1N1)) segment 8 nuclear export protein (NEP) and nonstructural protein 1 (NS1) genes, complete cds | https://www.ncbi.nlm.nih.gov/nuccore/MT666232 | NCBI GenBank, MT666232 |
| Ryt-Hansen P, Krog JS, Breum SO, Hjulsager CK, Pedersen AG, Trebbien R, Larsen LE | 2021 | Influenza A virus (A/swine/Denmark/2016_17664_1p2/2016 (H1N1)) segment 2 polymerase PB1 (PB1) and PB1-F2 protein (PB1-F2) genes, complete cds | https://www.ncbi.nlm.nih.gov/nuccore/MT666233 | NCBI GenBank, MT666233 |

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
