## [Decision Letter]

**Acceptance summary:**

The influenza A pandemic in 2009 has led to an increase in surveillance for swine influenza A virus. The authors describe the results of the surveillance of this virus in Danish swine from 2011 to 2018. Based on a substantial sample size the dynamics of sub- and genotype evolution are analysed. A newly emerging cluster of genetically drifted viruses has been identified. Viral fitness, modulated by frequent reassortments, is pointed out as a potentially major driver of swIAV evolution.

**Decision letter after peer review:**

Thank you for submitting your article "Co-circulation of multiple influenza A variants in swine harboring genes from seasonal human and swine influenza viruses" for consideration by *eLife*. Your article has been reviewed by 3 peer reviewers, and the evaluation has been overseen by a Senior/Reviewing Editor. The reviewers have opted to remain anonymous.

The reviewers have discussed the reviews with one another and the Reviewing Editor has drafted this decision to help you prepare a revised submission.

Phylogenetic analysis of swIAV viruses obtained in Denmark between 2011-2018 through passive surveillance is conducted on basis of a substantial sample size. Dynamics of sub- and genotype evolution are analysed and a newly emerging cluster of genetically drifted H1pdm viruses has been identified. Viral fitness, modulated by frequent reassortments, is pointed out as a potentially major driver of swIAV evolution. This is a very well conducted study providing timely, interesting data that widen the view on influenza virus evolution in a major mixing vessel species. The data come from a country which acts a main motor of swine production and trade in Europe.

The breadth and presentation of the data however is very difficult to follow.

Essential revisions:

1] The study entirely concentrates on production and analysis of genetic data.

However, antigenic analyses are sorely missed since in fact the data shown and discussed suggest important implications regarding antigenic drift. Therefore, some of the central conclusions seem to be weakly backed by the data:

L356-360: "Internal gene cassettes as drivers of viral evolution instead of population immunity." To support this hypothesis antigenic distances of the H1av proteins of the different reassortants of H1avN2dk would have been highly informative.

L366: "No immunity against the HA protein in the human population." Again, not studied here. In contrast, previous studies (to be cited: PMIDs 32601207 and 32721380) show substantial serologic (cross) reactivity in adult human sera against a number of swIAV isolates with increased antigenic distances to viruses circulating in the human population.

2] There is a misuse of the word "subtype" throughout the paper when phylogenetic clade or lineage is the correct influenza terminology. See for instance Line 30: Here and throughout, do you have 10 subtypes? Should this be 10 phylotypes or HA/NA genotypes?

When strain names are shown, they must be in correct influenza nomenclature format and with subtype included.

Thus, the authors should thoroughly re-check the paper for consistency of the nomenclature.

The use of abbreviated lineages of HA and NA is hard to follow.

3] There are major problems with the phylogenetic analyses, including too few swine and human IAV reference sequences and lack of a description of the methods for sampling reference data, and poor quality of trees used as figures. The other sequence analyses also lack enough additional reference sequences and clear description of what data was included.

4] The authors should be more cautious regarding the "zoonotic potential" of swIAV and to change the impact statement and some subtitles. Knowledge of amino acid substitutions that may enable adaptation of swine-adapted influenza viruses to humans is almost non-existent (see also Pulit-Penaloza et al., Trop Med Infect Dis. 2019;4(41);1-21).

Moreover, it remains unknown why the 2009 pandemic H1N1 influenza virus had the capacity to become adapted to humans, unlike other swine-origin influenza viruses. To our knowledge, most of the "specific host markers" mentioned are based on studies with wholly avian influenza viruses of various HA subtypes in mammalian cell culture or in ferrets, and not on studies with swine influenza viruses in humans. Therefore, the reviewers do not agree that "the zoonotic potential (was) evaluated (impact statement)". This would require a completely different approach. The authors themselves also conclude that "No unique swine or human residues were revealed.…" (line 194-196) and "none of the aa differences were 100% specific to each group" (line 300-1).

5] Similar comments apply to the paragraph about MxA resistance (lines 488-492). It has been suggested by one research group, and based on in-vitro studies, that MxA resistance may increase the zoonotic potential of animal influenza viruses, but it has never been firmly proven. For that purpose, one would need studies in human volunteers with MxA-resistant and non-resistant viruses. We do not agree with the statement that "MxA resistance is essential for zoonotic and pandemic potential of avian and swine IAV" (lines 490-492) and lines 491-492.

6] The paper is overly long and, as said above, the breadth and presentation of the data is very difficult to follow. This needs attention. For instance, the first two paragraphs of the Results section (Field samples – SwIAV positive samples) could be combined and reduced.

7] The discussion is also too long, which is due to redundancies of result descriptions. Streamlining might be achieved by removing large parts of L395-405, L423-430, L435-466.

8] There are many figures and tables, the need for so many is not clear. In addition, the titles and legends do not support the figures or tables standing on their own. We suggest merging and improving tables/figures to make the main points of the manuscript. Further streamlining would include removal of Table 1 to the supplements. Figures 2 and 4 should be combined while 1 and 3 might be supplemental. It is not clear why two figures of H1pdm phylogenetics are shown; Figure 7 should be sufficient. Figure 9 as well might be moved to the supplements.

---

## [Author Response]

Essential revisions:1] The study entirely concentrates on production and analysis of genetic data.However, antigenic analyses are sorely missed since in fact the data shown and discussed suggest important implications regarding antigenic drift. Therefore, some of the central conclusions seem to be weakly backed by the data:L356-360: "Internal gene cassettes as drivers of viral evolution instead of population immunity." To support this hypothesis antigenic distances of the H1av proteins of the different reassortants of H1avN2dk would have been highly informative.

Thank you for noticing and suggesting this. A sentence has now been added to consider the lack of antigenic investigations performed in this study and the sentence has been rephrased.

L366: "No immunity against the HA protein in the human population." Again, not studied here. In contrast, previous studies (to be cited: PMIDs 32601207 and 32721380) show substantial serologic (cross) reactivity in adult human sera against a number of swIAV isolates with increased antigenic distances to viruses circulating in the human population.

This sentence has now been rephrased and the two studies cited.

2] There is a misuse of the word "subtype" throughout the paper when phylogenetic clade or lineage is the correct influenza terminology. See for instance Line 30: Here and throughout, do you have 10 subtypes? Should this be 10 phylotypes or HA/NA genotypes?When strain names are shown, they must be in correct influenza nomenclature format and with subtype included.Thus, the authors should thoroughly re-check the paper for consistency of the nomenclature.The use of abbreviated lineages of HA and NA is hard to follow.

The word subtype has now been corrected to lineages and strains. And the naming of the different lineage has been checked for consistency and the suggestions for the abbreviations mentioned below has been included. However the abbreviated lineages of the HA and NA are needed to understand the origin of the novel ressortants and are also used similarly in recent papers describing the European swIAV (https://doi.org/10.1016/j.chom.2020.07.006 and https://journals.asm.org/doi/10.1128/JVI.00988-18).

3] There are major problems with the phylogenetic analyses, including too few swine and human IAV reference sequences and lack of a description of the methods for sampling reference data, and poor quality of trees used as figures. The other sequence analyses also lack enough additional reference sequences and clear description of what data was included.

The phylogenetic analysis of the H1av and the H1pdm have now been updated to contain several reference sequences, and a section in the method describes how these were selected. As for the phylogenetic analysis of the remaining genes it was not the scope to investigate the differences to other foreign sequences. Therefore, these sequences were only compared to the Danish sequences obtained in the surveillance, which also provides the possibility of estimating genetic changes over time, as precise dates for sampling were available.

4] The authors should be more cautious regarding the "zoonotic potential" of swIAV and to change the impact statement and some subtitles. Knowledge of amino acid substitutions that may enable adaptation of swine-adapted influenza viruses to humans is almost non-existent (see also Pulit-Penaloza et al., Trop Med Infect Dis. 2019;4(41);1-21).Moreover, it remains unknown why the 2009 pandemic H1N1 influenza virus had the capacity to become adapted to humans, unlike other swine-origin influenza viruses. To our knowledge, most of the "specific host markers" mentioned are based on studies with wholly avian influenza viruses of various HA subtypes in mammalian cell culture or in ferrets, and not on studies with swine influenza viruses in humans. Therefore, the reviewers do not agree that "the zoonotic potential (was) evaluated (impact statement)". This would require a completely different approach. The authors themselves also conclude that "No unique swine or human residues were revealed.…" (line 194-196) and "none of the aa differences were 100% specific to each group" (line 300-1).

Thank you for your comment on this. We have now rephrased several statements regarding the zoonotic potential.

5] Similar comments apply to the paragraph about MxA resistance (lines 488-492). It has been suggested by one research group, and based on in-vitro studies, that MxA resistance may increase the zoonotic potential of animal influenza viruses, but it has never been firmly proven. For that purpose, one would need studies in human volunteers with MxA-resistant and non-resistant viruses. We do not agree with the statement that "MxA resistance is essential for zoonotic and pandemic potential of avian and swine IAV" (lines 490-492) and lines 491-492.

This line has now been rephrased.

6] The paper is overly long and, as said above, the breadth and presentation of the data is very difficult to follow. This needs attention. For instance, the first two paragraphs of the Results section (Field samples – SwIAV positive samples) could be combined and reduced.

The paper have now been shortened and several of the table and figures deleted or moved to the supplements.

7] The discussion is also too long, which is due to redundancies of result descriptions. Streamlining might be achieved by removing large parts of L395-405, L423-430, L435-466.

The mentioned sections have been shortened and the discussion has been streamlined.

8] There are many figures and tables, the need for so many is not clear. In addition, the titles and legends do not support the figures or tables standing on their own. We suggest merging and improving tables/figures to make the main points of the manuscript. Further streamlining would include removal of Table 1 to the supplements. Figures 2 and 4 should be combined while 1 and 3 might be supplemental. It is not clear why two figures of H1pdm phylogenetics are shown; Figure 7 should be sufficient. Figure 9 as well might be moved to the supplements.

The tables and figures have now been revised, and several figure and tables deleted or moved to the supplements.